# Adherence to Telemonitoring by Electronic Patient-Reported Outcome Measures in Patients with Chronic Diseases: A Systematic Review

**DOI:** 10.3390/ijerph181910161

**Published:** 2021-09-27

**Authors:** Jim Wiegel, Bart Seppen, Marike van der Leeden, Martin van der Esch, Ralph de Vries, Wouter Bos

**Affiliations:** 1Amsterdam Rheumatology and Immunology Center, Reade, 1056 AA Amsterdam, The Netherlands; b.seppen@reade.nl (B.S.); m.vd.leeden@reade.nl (M.v.d.L.); m.vd.esch@reade.nl (M.v.d.E.); w.bos@reade.nl (W.B.); 2VU Medical Center, Department of Rheumatology, Amsterdam UMC, 1081 HV Amsterdam, The Netherlands; 3VU Medical Center, Department of Rehabilitation Medicine, 1081 HV Amsterdam, The Netherlands; 4CoE Urban Vitality, Faculty Health, Amsterdam University of Applied Sciences, 1081 HV Amsterdam, The Netherlands; 5Medical Library, Vrije Universiteit Amsterdam, 1081 HV Amsterdam, The Netherlands; r2.de.vries@vu.nl

**Keywords:** adherence, patient reported outcomes, patient reported outcome measures, telemonitoring

## Abstract

*Background:* Effective telemonitoring is possible through repetitive collection of electronic patient-reported outcome measures (ePROMs) in patients with chronic diseases. Low adherence to telemonitoring may have a negative impact on the effectiveness, but it is unknown which factors are associated with adherence to telemonitoring by ePROMs. The objective was to identify factors associated with adherence to telemonitoring by ePROMs in patients with chronic diseases. *Methods:* A systematic literature search was conducted in PubMed, Embase, PsycINFO and the Cochrane Library up to 8 June 2021. Eligibility criteria were: (1) interventional and cohort studies, (2) patients with a chronic disease, (3) repetitive ePROMs being used for telemonitoring, and (4) the study quantitatively investigating factors associated with adherence to telemonitoring by ePROMs. The Cochrane risk of bias tool and the risk of bias in nonrandomized studies of interventions were used to assess the risk of bias. An evidence synthesis was performed assigning to the results a strong, moderate, weak, inconclusive or an inconsistent level of evidence. *Results:* Five studies were included, one randomized controlled trial, two prospective uncontrolled studies and two retrospective cohort studies. A total of 15 factors potentially associated with adherence to telemonitoring by ePROMs were identified in the predominate studies of low quality. We found moderate-level evidence that sex is not associated with adherence. Some studies showed associations of the remaining factors with adherence, but the overall results were inconsistent or inconclusive. *Conclusions:* None of the 15 studied factors had conclusive evidence to be associated with adherence. Sex was, with moderate strength, not associated with adherence. The results were conflicting or indecisive, mainly due to the low number and low quality of studies. To optimize adherence to telemonitoring with ePROMs, mixed-method studies are needed.

## 1. Introduction

The increasing incidence of chronic diseases and the proportional increase in health care costs require efficacy in healthcare [1,2,3]. To improve efficacy, new ways of delivering healthcare were evaluated, such as the use of telemonitoring for chronic diseases. Telemonitoring, or remote patient monitoring, is defined as the use of technology to monitor patients at a distance [4]. One way of telemonitoring is collecting repetitive electronic patient-reported outcome measures (ePROMs) at home, which allows for a quick and easy way of a frequent capture of important disease-specific outcomes and is already applied in a variety of chronic diseases such as chronic obstructive pulmonary disease, chronic kidney disease, rheumatoid arthritis, inflammatory bowel disease or congestive heart failure [5,6,7,8,9,10]. The benefits of telemonitoring by repetitive ePROMs are (i) a reduced number of outpatient visits, (ii) a reduced workload, and (iii) lower healthcare costs [11,12,13]. Patients that use repetitive ePROMs for telemonitoring reported (iv) improved satisfaction, (v) improved communication with the provider, (vi) more insight into their disease activity, and (vii) more control over their disease [14,15,16]. However, one of the main problems of telemonitoring by ePROMs is a lack of adherence, with adherence rates down to 20% [17,18]. Although adherence toward health technology is increasingly investigated, there is no comprehensive review regarding what quantitative factors are associated with adherence to telemonitoring by ePROMs specifically. 

The low rate of adherence may affect the effectiveness of ePROMs. It is important to identify which factors affect adherence to telemonitoring by repetitive ePROMs, since this allows for a targeted approach in order to improve adherence and therefore its effectiveness [19]. A recent systematic review qualitatively investigated patients’ experience of telemonitoring (including but not limited to telemonitoring with ePROMs) and suggested that older patients and patients with less experience with technology were concerned about their ability to use telemonitoring [20]. Furthermore, the coexistence of comorbidity, social support, self-discipline and the use of persuasive design principles in the telemonitoring tool (i.e., use of reminders) were all qualitatively identified as possible factors affecting adherence [21,22,23,24]. However, it is not yet known if they influence adherence to telemonitoring by ePROMs specifically and if the quantitative evidence supports and extends these observations.

A systematic review was performed with the aim to identify factors quantitatively associated with adherence to telemonitoring by repetitive ePROMs in patients with chronic diseases in all studies. 

## 2. Materials and Methods

### 2.1. Search Strategy

A literature search was performed from inception to 6 June 2020 and was updated on 8 June 2021, based on the preferred reporting items for systematic reviews and meta-analyses (PRISMA) statement (www.prisma-statement.org, accessed on 5 March 2020). At the initiation of this review, the new PRIMSA guidelines were not yet available; therefore, this review was conducted following the 2009 PRISMA guidelines [25]. To identify all relevant publications, we conducted systematic searches in the bibliographic databases PubMed, Embase, PsycINFO (Ebsco) and the Cochrane Library (Wiley), in collaboration with a medical information specialist. The following terms were used (including synonyms and closely related words) as index terms or free-text words: “electronic patient reported outcome”, “e-Health”, “Telemonitoring”, “Remote patient monitoring”, “Mobile applications”, “Chronic disease”, “Adherence”, “Usage”, “Dropout”, Engagement” and “Compliance”. The references of the identified articles were searched for relevant publications. Duplicate articles were excluded. The full search strategies for all databases can be found in Appendix A (Table A1).

### 2.2. Selection Process

Two reviewers (JW and BS) independently screened all potentially relevant titles and abstracts for eligibility. We selected articles where (P) patients with chronic diseases (I) telemonitored their symptoms with repetitive ePROMs, (C) no comparison groups were necessary, and (O) where factors potentially positively or negatively associated with adherence were investigated, (S) in trial or cohort studies. Differences in judgement were resolved through consensus. Studies were included if they met the following criteria: Type of study is a randomized controlled trial (RCT), randomized controlled cross-over trial, clinical trial, prospective uncontrolled studies and cohort studies.Population consists of patients with a chronic diseaseRepetitive ePROMs are used for telemonitoringAdherence to reporting repetitive ePROMs is describedThe study quantitatively analyzes at least one factor potentially associated with adherence to telemonitoring by repetitive ePROMsWritten in Dutch or English

Exclusion criteria were: 

The described chronic disease is a mental disorder according to the *Diagnostic and Statistical Manual of Mental Disorders-5* [26].

If necessary, the full text article was checked for the presence of all inclusion criteria.

### 2.3. Data Collection Process and Data Items 

Data were extracted with the aid of a standardized data form by one reviewer (JW) [27]. The second reviewer (BS) sampled 50% of the articles for accuracy. The extracted information of each study consisted of: author, year of publication, study design, region, participant characteristics (sample size, diagnosis, age, sex, education level, work status and clinical characteristics), telemonitoring characteristics (design, frequency of intended usage, definition of adherence and automated reminders) and outcome measures (adherence rates and results for each investigated factor potentially affecting adherence). 

### 2.4. Methodological Quality of Individual Studies

Both researchers (JW and BS) assessed the risk of bias of the included studies independently. Disagreements were solved by discussion, and if necessary, a third party was consulted (WB). For RCTs the Cochrane risk of bias tool was used and for cohort studies the risk of bias in nonrandomized studies of interventions (ROBINS-1) [28,29]. The Cochrane risk of bias tool aims to disclose relevant information to the risk of bias within a fixed set of domains through signaling questions in randomized trials. The domains of bias investigated were (1) random sequence generation, (2) allocation concealment, (3) selective reporting, (4) other sources of bias, (5) blinding of participants and personnel, (6) blinding of outcome assessment, and (7) incomplete outcome data and can be judged as “high risk”, “low risk” or “unclear risk”, if the data are insufficient for judgement. The ROBINS-1 tool evaluates the risk of bias in nonrandomized studies of interventions such as cohort studies and is based on the Cochrane risk of bias tool. The domains of bias that were assessed were (1) confounding, (2) selection of participants, (3) classification of intervention, (4) deviation from intended intervention, (5) missing data, (6) measurement of outcomes, and (7) selection of the reported results. Through signaling questions, each domain was classified as “low risk”, “moderate risk”, “serious risk”, “critical risk” or “no information”. The overall risk of bias classification for the article was equal to the classification of the domain with the highest risk of bias. Articles describing subgroup analyses of previously performed RCTs were assessed as an RCT, and the original publication was retrieved to assess the methodological quality properly. 

### 2.5. Data Synthesis

The identified factors were categorized according to the WHO five dimensions associated with adherence, (1) social/economic factors (i.e., age and level of education), (2) health system/health care team-related factors (i.e., quality of consultations), (3) therapy related factors (i.e., treatment duration), (4) condition-related factors (i.e., symptom severity), and (5) patient related factors (i.e., self-efficacy) [30]. Although this categorization was intended for medication adherence or adherence to health therapy, we used the five dimensions for aggregating the results solely since such a classification does not exist for ePROM/technology adherence. If the factors were investigated in the included studies but were not described by the WHO, we categorized the factor in the most suitable dimension. To assess whether a meta-analysis was possible, we explored the heterogeneity of the included studies based on intervention (length of follow-up, app or web-based intervention, intended frequency of ePROMs and used description of adherence), context (diagnosis) and target participants (demographics, severity of symptoms and comorbidity) following the guidelines of Pigott et al. [31]. An evidence synthesis was performed when quantitative analysis was not possible due to high heterogeneity. In the evidence synthesis, the results were assigned to one of five levels of evidence: strong, moderate, weak, inconclusive or inconsistent following the criteria adapted from Ariëns et al.; see Table 1 [32]. When available, the results of univariable analysis were used for synthesis; otherwise, the results of multivariable analysis were used.

## 3. Results

The literature search generated a total of studies: 3943 in PubMed, 4339 in Embase, 714 in PsycINFO and 2485 in the Cochrane Library. After removing the duplicates of references that were selected from more than one database, 7275 studies remained. All the remaining studies were screened based on title and abstract. A total of 51 full-text studies were assessed for eligibility, and finally five studies were included in the synthesis. The flow chart of the search and selection process is shown in Figure 1. 

The included studies consisted of one article that analyzed the subgroups of RCTs, two prospective uncontrolled studies and two retrospective cohort studies [33,34,35,36,37]. Study populations comprised of patients with rheumatoid arthritis, heart failure, chronic pain and systemic auto-immune diseases. Publication dates varied between 2013 and 2020; see Table 1. A detailed description of the study characteristics is presented in Table 2. Table 3 shows the tool and intervention characteristics. 

The RCT was judged at high risk of bias on one domain, blinding of participants and personnel [33]. Both prospective uncontrolled studies were judged at serious risk due to risk of confounding [34,35]. Both retrospective cohort studies were judged at moderate risk [36,37]. The full quality assessment is presented in Figure 2 for RCTs and Figure 3 for prospective uncontrolled and retrospect cohort studies. The heterogeneity of the studies was considered too high to perform a meta-analysis; therefore, an evidence synthesis was performed.

### 3.1. Adherence

Overall adherence to repetitive ePROMs ranged between 61% and 96% in the studies. The definition of adherence varied between studies. In three studies, the percentage of completed questionnaires was used as a proxy for adherence and reported as a continuous variable, and two studies had a predefined number as cut-off point for high/low.

### 3.2. Factors Affecting Adherence Classified by the Five WHO Dimensions

A total of five studies investigated 15 unique factors. Of the factors investigated, eight belonged to the social/economic dimension (1), two factors to condition related (2), three to the patient related dimension (3), one to the therapy related dimension (4), and one to the healthcare team related dimension (5); see Table 4.

### 3.3. Social/Economic Dimension

There was moderate-level evidence that sex is not associated with adherence, as none of the five studies found sex was associated with adherence. The evidence for the association of age and marital status with adherence was inconsistent, while education level, annual income, number of children, area of residence and employment status were inconclusive. Colls et al. showed that patients over 65 years old had higher adherence compared to patients <45 years old, 77% vs. 62% *p* = 0.02 (32). By contrast, the four other studies did not find any difference regarding age and adherence [34,35,36,37]. Jamilloux et al. found an association between higher response rate and being married/having a partner (OR = 2.89, *p* = 0.02), in contrast to two studies who did not find any difference in adherence rates regarding marital status [34,35,36]. Higher education, annual income, area of residence (urban or not), employment status and number of children at home were all investigated by a single study, of which only a greater number of children at home was associated with higher adherence (average of 1.1 children at nonadherent group vs. 1.3 children at adherent group *p* = 0.004) [34]. 

### 3.4. Condition-Related Dimension

There was inconsistent and inconclusive evidence that symptom severity and disease duration, respectively, affected adherence. Colls et al. found that lower baseline disease activity, or lower symptom severity, was associated with higher adherence in comparison with higher disease activity (76% vs. 58%, *p* = 0.02), in contrast to Ross et al. who found that a higher symptom severity was associated with more daily ePROM reports [33,37]. The association between symptom duration and adherence was only investigated by Ross et al., who did not find an association. 

### 3.5. Patient-Related Dimension

The coexistence of comorbidity (including depression) and previous participation in online surveys was investigated in only one study [36]. Guzman et al. showed that a higher score on the Charlson comorbidity index was correlated with lower adherence (beta = −0.13, *p* < *0*.05) (35). Jamilloux et al. used previous participation in online surveys as surrogate for eHealth literacy and found it to be associated with higher adherence (61% in the adherent group vs. 38% in the nonadherent group OR = 2.56 95% CI 1.02–6.72 *p* = 0.04) [34]. According to Rosen et al., depression was not associated with adherence [35].

### 3.6. Therapy/Intervention-Related Dimension

Evidence for the association of satisfaction with the app/web-based intervention and adherence was inconsistent. Ross et al. showed that the number of daily pain entries was positively associated with satisfaction of the app, while Colls et al. did not find an association between global treatment satisfaction and adherence [33,37].

### 3.7. Healthcare-Team-Related Dimension

Guzman et al. found a positive association between higher adherence and patients treated at primary care, compared to patients treated at specialized care (OR 2.51 95%CI 1.22–5.19 *p* < 0.03) [36]. Since this was investigated by only one study, the result was classified as inconclusive.

## 4. Discussion

One of the main problems of telemonitoring by ePROMs is a lack of adherence and the lack of knowledge of the factors that influence adherence to telemonitoring. This review aimed to identify which factors are associated with adherence to telemonitoring by ePROMs in patients with chronic diseases. The definition of adherence varied between studies, with reported adherence ranging from 61% to 96%. Symptom severity, comorbidity, marital status, eHealth literacy and satisfaction with the intervention may be associated with higher adherence, but the evidence was not conclusive for any of the identified factors. Moderate evidence was found that sex is not associated to adherence.

We quantitatively investigated adherence to telemonitoring with ePROMs, in addition to previously performed qualitative research. Recent qualitative research toward telemonitoring (including but not limited to telemonitoring by ePROMs) found that coexistence of comorbidity, older age and eHealth illiteracy may negatively affect adherence [20,22]. In addition, increased social support, higher self-discipline and the usage of persuasive design in the telemonitoring tool were all qualitatively identified as possible factors positively affecting adherence [21,23,24]. Three of these factors were investigated quantitatively in our included studies: older age, eHealth literacy and comorbidity.

Colls et al. reported higher adherence in older patients, which is contrary to the earlier mentioned qualitative research and the widely accepted unified theory of acceptance and use of technology (UTAUT2) framework [38]. The UTAUT2 describes that older people may have more difficulty in adapting new technologies and hypothesized that older people are less frequently eHealth literate. However, a recent study in patients with chronic diseases showed that only eHealth literacy, and not age, was predictive for eHealth adherence [39]. The fact that smartphone usage in 55+ years old increased from 40% in 2014 to 90% in 2017 in the Netherlands suggests that the eHealth illiterate group might be decreasing in patients of older age [40]. Therefore, age seems to be of diminishing importance in eHealth adherence, while eHealth illiteracy remains a potential barrier. Guzman et al. identified the presence of comorbidity as a factor negatively affecting adherence, in accordance with qualitative research [22]. Multimorbidity is frequently present in patients with chronic diseases, for example in RA with an estimated prevalence of two-thirds of the patients [41]. Described rationales for this negative impact on adherence are that (1) comorbidity may physically limit the use of mHealth, and (2) it may shift the patients’ priority away from the primary disease [23]. Therefore, patients with multimorbidity are potentially in extra need for attention or help in order to achieve higher adherence. Measures such as the index of coexistent diseases are best-suited to study the impact of comorbidity on adherence, since they include both the disease severity and physical impairment of the comorbidities [42,43].

Symptom severity was not identified by qualitative research as a barrier for telemonitoring adherence but showed remarkable contradictive results in the included studies, acting both as a facilitator and barrier [33,37]. In studies investigating medicine adherence, it has mainly been described as a potential barrier. Patients considered high symptom severity as a barrier (i) when they are becoming skeptical toward the efficacy if previous treatment had failed, (ii) when establishing routines for disease management are difficult when symptoms are severe, (iii) and when patients experience a reduced quality of interaction with their healthcare provider when symptoms are severe [44,45,46]. However, high symptom severity may act as a facilitator if the patients experience greater benefits from the intervention when the symptoms are severe [37,47]. Therefore, the role of symptom severity on adherence may depend on the patients’ perceived benefits of telemonitoring and how easy the tool is to use even when symptom severity is high.

Only one factor was investigated in both the intervention-related and healthcare-related dimension. However, investigating intervention and healthcare factors may be valuable. For example, a high level of assistance of the healthcare practitioner may have a positive influence on adherence [48,49]. Furthermore, the frequency at which patients need to report ePROMs influences adherence is unknown. A higher frequency may lead to reporting fatigue and therefore lower adherence, but a higher frequency may also facilitate adherence in acquiring the habit more easily. Future studies investigating ePROM adherence should focus more on healthcare- and intervention-related dimensions. Not many factors are investigated both qualitatively and quantitatively, mainly due to the low number of studies investigating adherence of telemonitoring with ePROMs quantitatively. Quantitative research may help to identify which factors lead toward a decrease or increase in adherence on a group level. Furthermore, it gives the possibility of monitoring the effect of adjustments for improving adherence on group level. Therefore, it is of importance to combine qualitative and quantitative studies in prospective mixed-method studies in order to investigate thoroughly the adherence of telemonitoring by ePROMs.

Given the presented results, where the required skills and ease of use of the telemonitoring tool were mentioned as possible reasons why factors such as eHealth illiteracy, higher disease state and comorbidity may negatively impact adherence, it seems logical to design and use telemonitoring tools with a highly perceived ease of use as a possible first step to improve adherence. It is already known that this is an important factor in the successful implementation of eHealth; however, it also seems to be of importance for the sustained usage over time for telemonitoring [50]. This emphasizes the importance of a user-centered design [51].

### Limitations

This review has several limitations which should be considered when interpreting the results. First, comparing results of the studies is difficult because of the large heterogeneity in diagnosis, intervention and outcome measurements between the included studies. This is partly due to the inclusion of all chronic diseases which was necessary since studies investigating ePROM adherence quantitatively are scarce, despite the increasing interest in telemonitoring by ePROMs especially during the COVID-19 pandemic. During the pandemic, patients with chronic diseases were often instantly telemonitored since outpatient clinic visits limited their access to high-priority care only. Therefore, there must be many currently unpublished data, which can help us understand which patients need extra assistance or are better suitable for telemonitoring. We hope that by underlining this knowledge gap, we can motivate others to analyze and publish their adherence data. Secondly, the methodological quality of the studies was low, leading to weak, inconsistent or inconclusive evidence. This underlines the need for large, well-designed prospective cohort studies with the primary focus on investigating adherence. Thirdly, there is a persisting general lack of knowledge about how often ePROMs should be reported in order for telemonitoring to be effective [52]. This makes determining adherence mostly arbitrary.

## 5. Conclusions

The factor sex is not associated with adherence to repetitive ePROMs. Although several studies reported various associations between factors and adherence, the results show to be inconsistent or inconclusive. A limited number of telemonitoring studies by ePROMs report in-depth adherence data, making it difficult to draw conclusions from the studied factors. Future ePROM-guided telemonitoring studies should report adherence data in a standardized way. Mixed-method analyses are preferred for studies investigating telemonitoring adherence with ePROMs.

## Figures and Tables

**Figure 1 ijerph-18-10161-f001:**
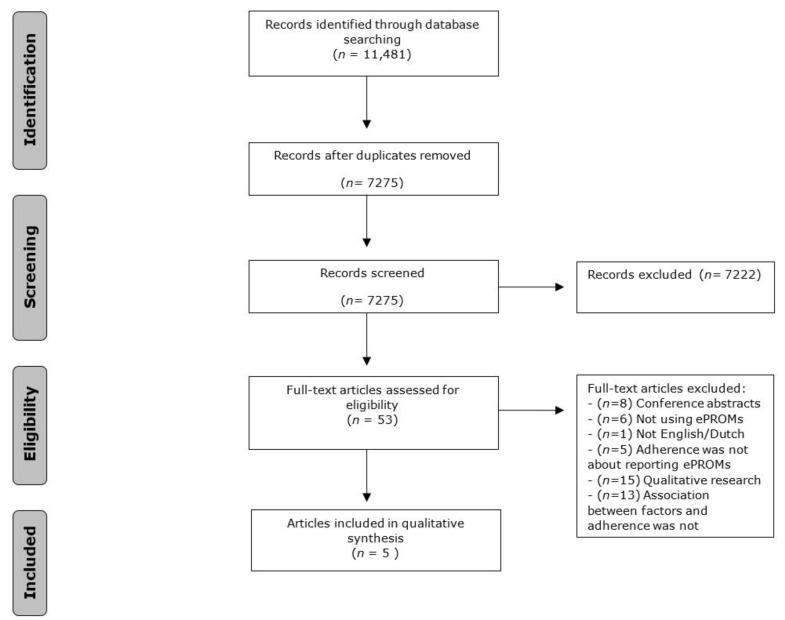
Flowchart of the search and selection process.

**Figure 2 ijerph-18-10161-f002:**
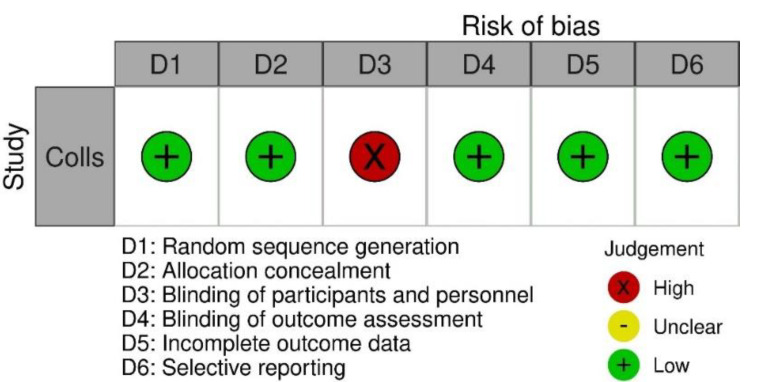
Risk of bias assessment following the Cochrane risk of bias tool for RCTs.

**Figure 3 ijerph-18-10161-f003:**
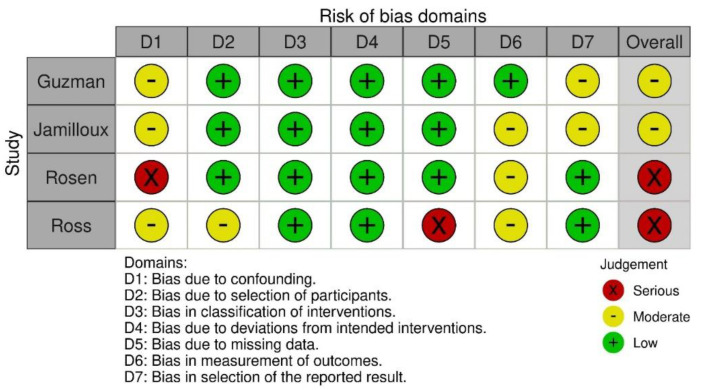
Risk of bias assessment following the ROBINS-1 for nonrandomized interventions.

**Table 1 ijerph-18-10161-t001:** Strength of evidence criteria [32].

Strength of Evidence	Criteria
Strong	At least 2 high-quality studies with consistent findings
Moderate	1 high-quality study and at least 2 low-quality studies with consistent findings
Weak	At least 2 low-quality studies with consistent findings
Inconclusive	Insufficient or conflicting studies
Inconsistent	Agreement of findings <75% of studies

**Table 2 ijerph-18-10161-t002:** Study characteristics of the included studies.

Author	Study Design	Year	Disease	Study Duration	Sample Size	Age, Mean (SD)	Men (%)
Colls J. et al.	RCT	2020	Rheumatoid Arthritis	6 months	78	55 (10.7)	15 (19%)
Jamilloux Y. et al.	Prospective uncontrolled study	2015	SLE, primary Sjögren and IBD	6 months	128	42 (median)	27 (21%)
Rosen D. et al.	Prospective uncontrolled study	2017	Heart failure	4 months	50	61 (-)	14 (29%)
Guzman, J.R.S. et al.	Retrospective cohort study	2013	Heart failure	3 months	248	76 (7.1)	240 (97%)
Ross, E.L. et al.	Retrospective cohort study	2020	Chronic pain	3 months	253	51 (14)	71 (28%)

* SLE = Systemic Lupus Erythematosus, IBD = Inflammatory Bowel Disease, COPD = Chronic Obstructive Pulmonary Disease.

**Table 3 ijerph-18-10161-t003:** Tool and adherence characteristics.

Authors	Tool Medium	Intended Frequency of Reporting	Definition of Adherence	Overall Adherence
Colls J. et al.	App or tablet	Daily	% completed	79%
Jamilloux Y. et al.	Web-based	Monthly	Good: 5 or 6 reported ePROMs Bad: <5	82%
Rosen D. et al.	Tablet	Daily	% completed	96%
Guzman, J.R.S. et al.	Tablet	Daily	<80% low >80% high adherence	61%
Ross. E.L. et al.	App	Daily	% of completed	69%

**Table 4 ijerph-18-10161-t004:** Overview of the investigated factors and their association with adherence.

		Colls J. et al.	Jamilloux Y. et al.	Rosen D. et al.	Guzman J.R.S. et al.	Ross E.L. et al.	Strength of Evidence
Social/economic dimension	Sex	↔	↔	↔	↔	↔	Moderate
	Increasing age	↑	↔	↔	↔	↔	Inconsistent
	Higher education	↔					Inconsistent
	Married		↑	↔	↔		Inconsistent
	Annual income				↔		Inconclusive
	Number of children at home		↑				Inconclusive
	Area of residence		↔				Inconclusive
	Employment status		↔				Inconclusive
Condition related dimension	Sever symptoms	↓				↑	Inconsistent
	Longer symptom duration					↔	Inconclusive
Patient related dimension	Depression			↔			weak
	Comorbidity				↓		Inconsistent
	Existing experience in online surveys		↑				Inconclusive
Therapy/intervention related dimension	Satisfaction with the app/web-based intervention	↔				↑	Inconsistent
Healthcare team related dimension	Primary care compared to specialized care				↑		Inconclusive

↑ = significant positive association with adherence, ↓ = significant negative association with adherence,
↔ = no significant association with adherence.

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
