# Peer review of "Adherence to Telemonitoring by Electronic Patient-Reported Outcome Measures in Patients with Chronic Diseases: A Systematic Review"

_ijerph, 2021, doi:10.3390/ijerph181910161_

Round 1
Reviewer 1 Report
Thank you for the opportunity to review this manuscript titled “Factors associated with adherence to telemonitoring with electronic patient-reported outcome measures in patients with 3 chronic diseases: a systematic review”. At this point in time, this study has few limitations. A few suggestions to make this report even better:
The title is complex. A modification is suggested to attract the reader's attention.
Line 18-19: “Identifying adherence related factors gives the possibility to deliberately improve adherence and subsequently the effectiveness of telemonitoring” The authors repeatedly use the term adherence, making the phrase redundant.
Line 20-21: “The objective is to identify factors associated with adherence to telemonitoring with ePROMs in patients with chronic diseases.” Authors have already completed their systematic review, so they should use past tense in their objective.
Line 22: Embase.com is the site web, please use “Embase”
Line 21-25. Please, include study eligibility criteria, study appraisal and synthesis methods.
Line 26. “RCT” Before you abbreviate, you have to spell out the full term in this section.
Line 30. The main aim of a conclusion is to provide an answer/resolution to the research question posed. Please, answer to your objective.
Line 34-35: Key words should be placed in alphabetical order.
Line 40-41. What is telemonitoring or remote patient monitoring? Is the same that telehealth or telemedicine? A description of the concept/term is necessary to understand the relevance of this manuscript.
Line 23-54. Perhaps it is a very forceful phrase.
Albert, N. M., Dinesen, B., Spindler, H., Southard, J., Bena, J. F., Catz, S., ... & Nesbitt, T. S. (2017). Factors associated with telemonitoring use among patients with chronic heart failure. Journal of telemedicine and telecare, 23(2), 283-291.
Alghamdi, S. M., Janaudis-Ferreira, T., Alhasani, R., & Ahmed, S. (2019). Acceptance, adherence and dropout rates of individuals with COPD approached in telehealth interventions: a protocol for systematic review and meta-analysis. BMJ open, 9(4), e026794. https://doi.org/10.1136/bmjopen-2018-026794
Kamei, T., Kanamori, T., Yamamoto, Y., & Edirippulige, S. (2020). The use of wearable devices in chronic disease management to enhance adherence and improve telehealth outcomes: A systematic review and meta-analysis. Journal of Telemedicine and Telecare, 1357633X20937573.
Sanders, C., Rogers, A., Bowen, R., Bower, P., Hirani, S., Cartwright, M., ... & Newman, S. P. (2012). Exploring barriers to participation and adoption of telehealth and telecare within the Whole System Demonstrator trial: a qualitative study. BMC health services research, 12(1), 1-12.
Walker, R. C., Tong, A., Howard, K., & Palmer, S. C. (2019). Patient expectations and experiences of remote monitoring for chronic diseases: systematic review and thematic synthesis of qualitative studies. International journal of medical informatics, 124, 78-85.
Line 72-74. Please, Check the latest update. The review protocol should have been recorded (https://www.crd.york.ac.uk/prospero/)
Page, M. J., McKenzie, J. E., Bossuyt, P. M., Boutron, I., Hoffmann, T. C., Mulrow, C. D., ... & Moher, D. (2021). The PRISMA 2020 statement: an updated guideline for reporting systematic reviews. Bmj, 372.
Line 83. Please provide an explicit statement of questions being addressed with reference to participants, interventions/exposure, comparisons, outcomes, and study design.
Line 149. Maybe authors may revised their systematic search. Plee see below:
Jenkins, C., Burkett, N. S., Ovbiagele, B., Mueller, M., Patel, S., Brunner-Jackson, B., ... & Treiber, F. (2016). Stroke patients and their attitudes toward mHealth monitoring to support blood pressure control and medication adherence. Mhealth, 2.
Richter, J. G., Nannen, C., Chehab, G., Acar, H., Becker, A., Willers, R., ... & Schneider, M. (2021). Mobile App-based documentation of patient-reported outcomes—3-months results from a proof-of-concept study on modern rheumatology patient management. Arthritis Research & Therapy, 23(1), 1-9.
Yount, S. E., Rothrock, N., Bass, M., Beaumont, J. L., Pach, D., Lad, T., ... & Cella, D. (2014). A randomized trial of weekly symptom telemonitoring in advanced lung cancer. Journal of pain and symptom management, 47(6), 973-989.
Line 232-313: Telemonitoring is of particular interest at this time of Covid19 pandemic. Although it is not within the scope of the present article, I think that it would add value to the Discussion if the authors add information about this topic.
Line 333: please references need major revision.
Reviewer 2 Report
Thank you for asking me to review this manuscript that presents a study with the aim of identifying factors associated with adherence to telemonitoring with ePROMs in patients with chronic diseases. Nevertheless, there are some comments:
-line 221: I think that there is an error in the following phrase: "depression was not associated with adherence al (34)."
-Have the authors take into account the frequency that patients have to register the PROMS? Because it is different to do daily or monthly, perhaps doing it more often makes the patient acquire the habit more easily, or on the contrary, the patient tires sooner.
-In spite of the methods and the results are very clear, the scare of articles included do not allow clear conclusions. In addition, there are very different patients and that does not make things easier.
-It is not clear to me why the authors focus only on PROMs when in most articles that use telemedicine, for example in COPD, the variables used are objective (heart rate, body temperature, oxygen saturation ...) In my opinion, it should be a good option to include more variables or more study designs (qualitative studies) to enlarge the number of included studies, and to obtain clear conclusions.
Round 2
Reviewer 1 Report
Dears Authors,
in my opinion your manuscript has improved its quality.
Congratulations.
Author Response
Dear reviewer,
We'd like to thank you once more for your throrough review and are delighted that the improvements were to your satisfaction.
Sincerely,
Jim Wiegel
Reviewer 2 Report
I think that the authors have clarified all my doubts about the main comments to the manuscript. Although there are not a lot of studies included, I consider this manuscript a good way to take the first step that helps other authors to carry out and to improve this type of articles in the future.
Author Response

(The authors gave the same response as above.)
